# CONTEXT-NER: CONTEXTUAL PHRASE GENERATION AT SCALE

## ABSTRACT

Named Entity Recognition (NER) has seen significant progress in recent years, with numerous state-of-the-art (SOTA) models achieving high performance. However, very few studies have focused on the generation of entities' context. In this paper, we introduce CONTEXT-NER, a task that aims to generate the relevant context for entities in a sentence, where the context is a phrase describing the entity but not necessarily present in the sentence. To facilitate research in this task, we also present the EDGAR10-Q dataset, which consists of annual and quarterly reports from the top 1500 publicly traded companies. The dataset is the largest of its kind, containing 1M sentences, 2.8M entities, and an average of 35 tokens per sentence, making it a challenging dataset. We propose a baseline approach that combines a phrase generation algorithm with inferencing using a 220M language model, achieving a ROUGE-L score of 27% on the test split. Additionally, we perform a one-shot inference with ChatGPT, which obtains a 30% ROUGE-L, highlighting the difficulty of the dataset. We also evaluate models such as T5 and BART, which achieve a maximum ROUGE-L of 49% after supervised finetuning on EDGAR10-Q. We also find that T5-large, when pre-finetuned on EDGAR10-Q, achieve SOTA results on downstream finance tasks such as Headline, FPB, and FiQA SA, outperforming vanilla version by 10.81 points. To our surprise, this 66x smaller pre-finetuned model also surpasses the finance-specific LLM BloombergGPT-50B by 15 points. We hope that our dataset and generated artifacts will encourage further research in this direction, leading to the development of more sophisticated language models for financial text analysis [1].

## 1 INTRODUCTION

Recent advancements in Named Entity Recognition (NER) have led to impressive results through the development of various large-scale pretrained models Zhang et al. (2023); Ma et al. (2023); Zhang et al. (2022); Yuan et al. (2021); Wang et al. (2021b;a). While existing research has primarily focused on NER task performance Zhang & Zhang (2022); Wang et al. (2014); Francis et al. (2019); Alexander & de Vries (2021); Wu et al. (2022); Wang & Wang (2022); Varshney et al. (2022); Shrimal et al. (2022), limited attention has been given to exploring contextual information associated with the identified entities within sentences. To illustrate this challenge, consider the sentence "The rent due today is $500." In this example, the question "What is $500?" can be answered as "rent due today." This task becomes particularly challenging when sentences are lengthy, lack explicit contextual cues, or contain multiple entities. For instance, referring to the second sentence in Table 1, discerning the context of "2.2 Million" as "Valuation allowance for loan servicing rights with impairments" solely based on the sentence becomes difficult (Asking the question "What is 2.2 million ?" using the sentence as the context). Notably, the exact phrase is not directly present in the sentence; however, it holds significant relevance in accurately describing the entity. Estimating the context of the entity is very difficult for a non expert human user as well (§. 6.4.) We pose the following question "How do you train models that could capture the intrinsic value of an entity in a sentence?"

We propose CONTEXT-NER, a task which involves to generate a concept description of the entity given a sentence, regardless of whether the concept is explicitly present in the sentence. We also

---

[1]Dataset, the script to generate it, baseline approach, ChatGPT evaluations, and finetuned models are freely available at https://anonymous.4open.science/r/edgar10q-dataset-144D/README.md

| Sentences | Entity | Type | Context of entity |
|---|---|---|---|
| As of June 30, 2019, the department store loans discussed above were 90 days or greater past due, as were $4.5 million of residential loans and a $36.2 million infrastructure loan with a carrying value of $29.2 million, net of a $7.0 million non accretable difference. | $29.2 Million | Money | Carrying amount of loans 90 days or more past due |
| There were impairments of $0.8 million for the three months ended June 30, 2020 and $2.2 million for the six months ended June 30, 2020. | $2.2 million | Money | Valuation allowance for loan servicing rights with impairments |

Table 1: Example sentences from the dataset with sampled entities (Column 2) and their context (Column 4). As can be seen from Column 2, the entities do not convey the entire picture by themselves and generating their relevant context becomes an important task to address.

introduce the EDGAR10-Q dataset [2] to initiate a systematic study of the task. The dataset comprises of quarterly and annual financial reports from publicly traded limited liability companies (LLCs). These reports are prepared by domain experts (financial analysts), ensuring highest quality of gold labels and contextual relevance. Table 1 shows some examples of the dataset from which it is evident that phrases are not always present in sentences and can be difficult to retrieve without adequate knowledge of the domain. EDGAR10-Q is one of the largest in the financial domain (1M sentences 2.78M entities) and consists of complex sentences that are not prevalent in benchmark datasets, posing a new challenge for SOTA models. The dataset has two unique qualities that make it particularly challenging. Firstly, the sentences are long and complex in nature; averaging approximately 35 tokens per sentence, surpassing the length of sentences typically encountered in the training of large language models (LLMs). Secondly, since this dataset is prepared from financial documents, they contain several numerical entities whose context can be difficult to extract by using just one sentence (§3).

We conduct various experiments in algorithmic (rule based), one-shot and supervised learning settings. We introduce a baseline method that leverages syntactic trees of the sentence to generate questions and find relevant phrases in the sentences (§4). The baseline yields an overall result of 27.59 ROUGE-L score . We also conduct zero shot, one shot, few shot evaluations on ChatGPT Brown et al. (2020) to get max 36.81% score. Responses of the baseline approach and ChatGPT are illustrated in Table 2. We also train

| Baseline Response | ChatGPT Response |
|---|---|
| infrastructure loan | Infrastructure loan carrying value |
| Impairment | Impairment expenses for three and six months ended June 30, 2020. |

Table 2: Responses of the baseline approach and ChatGPT for sample sentences present in Table 1 The responses highlight the difficulty to generate relevant context associated with entites.

T5's Raffel et al. (2020) different variants (T5, Tk-Instruct Wang et al. (2022b), and Flan T5 Chung et al. (2022)) and BART model Lewis et al. (2020) in a supervised manner to get 49% as the highest ROUGE-L score (§5.2). The low scores are identified as an area for further research to enhance the learning capabilities for such complex tasks.

We examine the effects of the generated artifacts using the dataset. Our findings reveal that T5 pre-finetuned on EDGAR10-Q outperforms vanilla T5 by 10.81 points and surpasses BloombergGPT 50B by 15.81 points on various downstream finance datasets (§6.3). Additionally, we provide a comparison between CONTEXT-NER and OpenIE to highlight the distinctions between these tasks. We explore the effect of instruction tuning on the T5 model using the EDGAR10-Q dataset, resulting in a performance improvement of 2% points. Lastly, our human evaluation case study, involving non-experts, yields a best score of 36%, further underscoring the need for continued research in this domain.

**Contributions:** (a) we introduce the task CONTEXT-NER, to generate contextual phrases for entities in sentences and associated EDGAR10-Q dataset created from financial reports; (c) we evaluate the dataset using the following methods: (c.1) we introduce a baseline approach which achieves a 27%

---

[2] Named after the Electronic Data Gathering, Analysis, and Retrieval system, which performs automated collection, validation, indexing, acceptance, and forwarding of submissions by companies and others who are required by law to file forms with the U.S. Securities and Exchange Commission (SEC)

ROUGE-L; (c.2) we evaluate the dataset in a one-shot setting via ChatGPT achieving 30% ROUGE-L; (c.3) we train different generative models in a supervised manner to get $\sim 50\%$ performance; (d) we perform a detailed analysis on following lines of enquiry (d.1) effect of pre-finetuning using EDGAR10-Q to achieve SOTA on several finance downstream tasks (d.2) qualitative comparison of CONTEXT-NER with OpenIE (d.3) explore the effect of instruction tuning for EDGAR10-Q.

## 2 NEED AND SIGNIFICANCE OF THE TASK

Consider the example from Table 1: "Impairments of $0.8 million for the three months ended June 30, 2020 and $2.2 million for the six months ended June 30, 2020." Estimating the meaning of 2.2 million as "Valuation allowance for loan servicing rights with impairments" remains challenging for non-experts (Asking the question "What is 2.2 million ?" using the sentence as the context) We also see that from Table 2, ChatGPT and baselines struggle, emphasizing the necessity for research. However, subject matter experts who created gold labels could understand it. This led to the question: How to transfer domain expert knowledge to an LLM ?

Inorder to capture intrinsic entity value in sentences, we introduce ContextNER task and its associated artifacts (large benchmark dataset and fine tuned models released to the community). Our objective is to bridge the gap between experts and models by infusing domain knowledge into the models that goes beyond what is explicitly present in the sentence. In our domain-specific document corpora, generating pertinent contexts linked to various financial Named Entities holds substantial value for professionals making critical decisions based on such reports. Through the creation of this task, we've recognized the following beneficial use cases:

- Credit Companies (Risk Assessment): Credit companies could leverage the contextual understanding of entities to perform more accurate risk assessments of other companies. This would enable them to evaluate the financial health and stability of businesses more effectively, leading to better-informed credit decisions.

- Hedge Funds (Investment Decisions): By analyzing the contextual relationships between companies and other market factors, hedge funds could refine their strategies, resulting in more favorable investment outcomes.

- Finance Journalism (Accurate Reporting): With contextual understanding, finance journalists would be empowered to extract precise and up-to-date information from Edgar reports, enabling them to produce more accurate and insightful articles and reports.

- Regulatory Compliance (Efficient Reporting): Understanding the intricacies of entities in the reports would enable compliance to fulfill reporting obligations more efficiently, ensuring adherence to regulatory requirements.

## 3 EDGAR10-Q DATASET

**Dataset Creation:** The EDGAR10-Q dataset was created by scraping quarterly (10-Q) and annual (10-K) reports from the years 2019, 2020, and 2021. Given the crucial role these meticulously prepared reports play in assessing the financial health of organizations, great care is taken in their curation to ensure accuracy and quality, leaving no room for oversight. To ensure standardization, all SEC filings undergo a tagging process where entities within sentences are labeled with corresponding Named Entity Recognition (NER) context labels, serving as high-quality gold labels for the dataset. We refer the reader to §A for more details. [3]

| Entity Types | Counts |
|---|---|
| Floating Values (monetary and percent) | 2.1M |
| number of Assets (Shares and Integers) | 425.8K |
| Ordinal Values | 16.8K |
| Dates | 195K |

Table 3: Distribution of different types of entities in the dataset.

**Dataset Description:** Table 3 shows the four types of entities, namely money, time (duration), percent, and cardinal values (pure, shares, and integer) present in

---

[3]Stanford NER tagger http://nlp.stanford.edu:8080/ner/ recognizes the aforementioned types as named entities. We follow their convention for the entity recognition.

| Dataset name | Docs | Sentences | Words | Entities |
|---|---|---|---|---|
| funsd Guillaume Jaume (2019) | 200 | NA | 31485 | 9743 |
| wikicoref Ghaddar & Langlais (2016) | 30 | 2229 | 59652 | 3557 |
| scierc Luan et al. (2018) | 500 | NA | NA | 8089 |
| med ment. Patil (2020) | 4392 | 42602 | 1176058 | 352496 |
| genia Fu et al. (2020) | 2000 | 18545 | 436967 | 96582 |
| conll 2003 Sang & Meulder (2003) | 1393 | 22137 | 301418 | 35089 |
| **EDGAR10-Q** | **18752** | **1009712** | **77400425** | **2780969** |

Table 5: Systematic comparison of EDGAR 10-Q with other benchmark NER dataset.

| # of Entities | Train | | Test | |
|---|---|---|---|---|
| | # of Sent. | Avg. S. Len | # of Sent. | Avg. S. Len |
| **1** | 381251 | 31.40 | 53567 | 30.26 |
| **2** | 261687 | 35.82 | 31985 | 35.18 |
| **3** | 86020 | 41.97 | 10425 | 41.93 |
| **4** | 38500 | 52.02 | 4309 | 54.23 |
| **5 +** | 16726 | 73.01 | 1530 | 85.98 |
| **Overall** | **784184** | **35.93** | **101816** | **34.85** |

Table 6: Train and test split statistics of the dataset for supervised learning experiments.

the data. Table 4 further elucidate data richness through paragraph and sentence level statistics. The average length of the label is 4.55 tokens highlighting that sufficiently long context phrases are used to describe the entity. The train and test set are of roughly equal difficulty in terms of sentences.

Table 5 compares this dataset with benchmark NER datasets and contains nearly 18.7K documents comprising 1M sentences. As it can be inferred from the table, the EDGAR-10Q dataset has nearly 650x more documents and nearly 780x more entities in comparison to the popular NER benchmark dataset - wikicoref Ghaddar & Langlais (2016). We observe that the EDGAR10-Q is the largest and richest in multiple parameters and a first-of-its-kind dataset in the financial domain. Table 6 highlights the train test split of the dataset.

The dataset is also divided according to the different number of entities present in a sentence. We see that as the number of entities increases, the average sentence length increase as well. Since this is a real-world dataset, sentences with 1 entity are most prevalent and comprise 49% of the dataset while sentences with 5+ entities consist of 2% of the dataset (more details present in Table 20 in §F) .

| Statistic | Values |
|---|---|
| Entities / sentence | 1.78 |
| Words / paragraph | 113.14 |
| Avg tokens / label | 4.55 |
| Words / sentence | 35.88 |

Table 4: Relevant sentence and paragraph wise statistics of the dataset that highlight the task's difficulty.

# 4 BASELINE APPROACH

---

**Algorithm 1:** Phrase Generation

**Input:** Sentence
**Output:** List of Phrases

```
1  Function noun_phrase(Sentence):
2      doc = sequence_of_token(Sentence)
3      phrase_list = []
4      for token in Doc:
5          ph = '  '
6          if token.head.pos in [Noun, Pronoun] and
              token.dep in [Object, Subject]:
7              for subtoken in token.children:
8                  if subtoken.pos is Adj or
                      subtoken.dep is Comp:
9                      ph += subtoken.text + '  '
10             if len(ph) is not 0:
11                 ph += token.text
12         if len(ph) is not 0 and ph doesnot have
              entities:
13             phrase_list.append(ph)
14     return phrase_list
```

---

We present a simple, yet efficient method to extract entities' descriptions from sentences, as shown in Figure 1. OpenIE is predicated on the idea that the relation (which is action verbs in most cases) is the central element of the extraction process, from which all other considerations flow. However, in many cases, the verb is not helpful, particularly in financial data. Consider the sentence: "Deferred revenue for 2020 is $20 billion." Like most financial records are of the form "is, was, be," etc., the verb "is" in this sentence is an auxiliary verb and does not describe any particular event or give any information about the entity. Our approach consists of a phrase generation algorithm that is in turn used for the creation of questions and fed into a transformer model for machine reading comprehension.

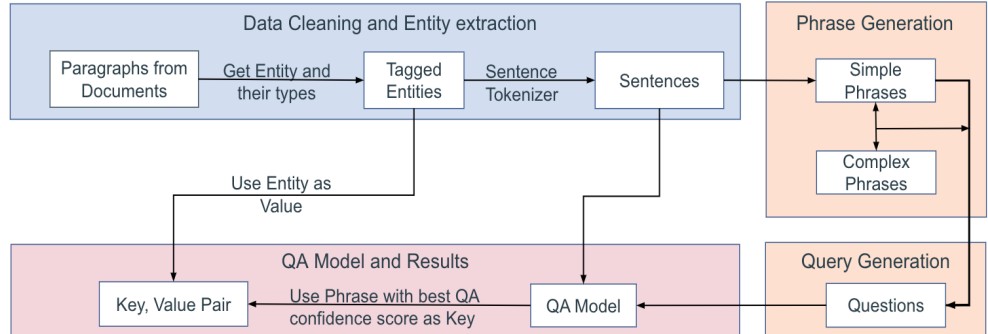

Figure 1: Illustration of the proposed baseline approach. Overall process using question generation and reading comprehension.

### 4.1 PHRASE EXTRACTION

A noun phrase (NP) is defined as phrase Stuart et al. (2013) containing a noun, a person, place, or thing, and the modifier that distinguishes it. We extract two types of phrases from the sentences, namely simple and complex. In simple phrase extraction, each sentence comprises subject-object and verb connecting them where the subject or object is usually a noun or pronoun.

After searching for a noun and pronoun, we check for any noun compound or adjective. On the other hand, for complex phrase extraction, we first start with preposition extraction. It has to be noted that simple phrases are not always found on both sides of the proposition. We then follow similar steps as in simple phrase extraction to look for phrases in both the left and right of the preposition. Consider an example,*In connection with the refinance we reduced the loan amount by $6.8 million.*. From our algorithm, the phrases extracted from this sentence are *loan amount* and *connection with refinance*. We use Spacy [4] library for POS tags of the word which were leveraged in Algorithm 1. We refer the reader to §C where the approach is described in more detail along with examples and flowchart (Algo. 2 and Fig. 4).

### 4.2 MACHINE READING COMPREHENSION FOR BASELINE

Phrases and entities are extracted on a sentence level for each paragraph. Based on the type of entity and the noun phrases, the questions are framed accordingly in a rule-based fashion. For instance, if the entity found out was of type date, then the question would be "when is" + NP?. Once these questions are generated, they are fed into the MRC Model, and the answers are checked for the relevant entity. If there are multiple questions with the same answer, we select the one with the highest confidence score.

There are instances where none of the generated questions returned an answer with the target entity or returned responses with a different entity. For those cases, we create the question "what is" entity? and its response would be considered as the relevant phrase.

In case these questions return different entities as responses, all cases to identify the noun phrase fail and the algorithm does not return a response. A detailed description of the MRC model is present in §C.2. Since this method does not require any finetuning, the baseline is directly evaluated on the test set.

| Evaluation Methods | P | R | F1 |
|---|---|---|---|
| **Baseline** | 34.59 | 27.06 | 27.59 |
| **ChatGPT zero shot** | 27.53 | 42.31 | 27.53 |
| **ChatGPT one shot** | 33.71 | 41.10 | 34.10 |
| **ChatGPT few shot** | 38.62 | 41.68 | 36.81 |

Table 7: Scores of different baseline approaches on test set of EDGAR 10-Q dataset. Baseline and the ChatGPT baselines give nearly the same performance.

---

[4]Spacy POS Tagging Library link: https://spacy.io/usage/linguistic-features.

| # of | BART Base | | | T5 Base | | | T5 Large | | | Flan T5 Base | | | Tk-Inst Large | | |
| Ent. | P | R | F1 | P | R | F1 | P | R | F1 | P | R | F1 | P | R | F1 |
|---|---|---|---|---|---|---|---|---|---|---|---|---|---|---|---|
| 1 | 50.26 | 45.98 | 45.71 | 49.33 | 44.91 | 44.50 | 50.44 | 46.38 | 45.76 | 49.46 | 45.09 | 44.65 | 49.44 | 45.18 | 44.66 |
| 2 | 54.44 | 51.03 | 50.47 | 53.65 | 50.30 | 49.45 | 54.96 | 51.89 | 50.94 | 53.96 | 50.49 | 49.71 | 53.81 | 50.36 | 49.54 |
| 3 | 54.74 | 51.56 | 50.89 | 53.31 | 49.84 | 48.95 | 55.16 | 51.82 | 50.93 | 53.74 | 49.97 | 49.31 | 53.71 | 50.01 | 49.30 |
| 4 | 54.37 | 51.36 | 50.69 | 53.86 | 50.29 | 49.49 | 55.47 | 52.60 | 51.66 | 53.55 | 50.25 | 49.37 | 53.60 | 50.29 | 49.42 |
| 5+ | 53.31 | 50.13 | 49.63 | 51.52 | 47.27 | 46.97 | 52.66 | 48.48 | 48.24 | 49.44 | 45.29 | 45.12 | 49.61 | 45.15 | 45.02 |
| Overall | 53.10 | 49.51 | **49.01** | 52.13 | 48.35 | **47.67** | 53.49 | 50.02 | **49.23** | 52.22 | 48.40 | **47.77** | 52.17 | 48.38 | **47.70** |

Table 8: Supervised learning scores of different models. P, R and F1 denote Precision, Recall and F1 respectively. 1,2,3,4,5+ denote the entity wise scores for different models.

### 4.3 CHATGPT EVALUATION

We establish three ChatGPT baselines that evaluate the test set in a zero-shot,one-shot and few-shot setting. This method is evaluated on 10% of the test set due to budget constraints. The prompts used for the experiment are present in §D.

## 5 EXPERIMENTS AND RESULTS

### 5.1 EXPERIMENTAL SETUP

**Baseline Model Setup:** We run all our experiments using the BERT base model Devlin et al. (2018). All experiments are done with Nvidia V100 16GB GPU.

**ChatGPT Setup:** We evaluate ChatGPT (gpt-3.5-turbo, max tokens = 256, top p = 1, frequency penalty = 0, presence penalty = 0) in zero,one and few shot setting.

**Performance Evaluation metrics:** ROUGE-L score uses the longest common subsequence matching between the baseline and GPT-3 responses to compare output quality. We report precision, recall, and the F1 measure against the ROUGE-L Lin (2004) score. We also report the Exact Match Rajpurkar et al. (2016) which measures the ratio of the instances for which a model's response has a ROUGE-l score of 1 with a gold label. We report No match where the generated output and gold label 0 ROUGE-L score.

### 5.2 RESULTS

**Baseline Scores:** Table 7 shows the baseline results where the overall F1 27.59%. Table 14 in §E gives the detailed results for the baseline.

**ChatGPT Scores:** Table 7 also shows Chat-GPT scores on the test set in zero-shot, one-shot and few-shot settings. As more examples are given to ChatGPT, the overall F1 increases. Precision of ChatGPT zero shot is lower than baseline, but recall is much higher, resulting in a higher overall F1. Detailed one-shot experiments on ChatGPT one-shot are present in Table 15.

**Supervised Training Results:** Table 8 shows the results when the generative models are fine-tuned with the train split and evaluated on test split. The overall supervised training performance is much higher as compared to baselines, where base models (T5, Flan, and BART Base) perform much better than ChatGPT (47.67%,

| Evaluation Methods | Exact Match Score | No Match Score |
|---|---|---|
| Baseline | 5.77 | 47.96 |
| ChatGPT (zero-shot) | 4.97 | 26.95 |
| ChatGPT (one-shot) | 3.67 | 26.37 |
| ChatGPT (few-shot) | 8.18 | 28.61 |
| Bart Base | 20.88 | 23.74 |
| T5 Base | 19.31 | 24.09 |
| T5 Large | 20.92 | 23.24 |
| Flan T5 Base | 19.64 | 24.24 |
| Tk Instruct Base | 19.44 | 24.23 |
| Tk Inst. w. Inst. | 21.45 | 22.83 |

Table 9: Exact Match and No Match scores for LLMs fine-tuned on EDGAR 10-Q dataset. On comparison with Baseline and ChatGPT results, we see that finetuning improves upon result quality.

| | Sentence | Entity | Labels | Baseline |
|---|---|---|---|---|
| | **Instances of Exact Match** | | | |
| S1 | Premium receivables are reported net of an allowance for doubtful accounts of $250 and $237 at September 30, 2020 and December 31, 2019, respectively. | $250 and $237 | premium receivable | premium receivable |
| S2 | The fair value of the collateral received at the time of the transactions amounted to $1,019 and $351 at September 30, 2020 and December 31, 2019, respectively. | $1,019 and $351 | fair value of collateral | fair value of collateral |
| | **Instances of No Match** | | | |
| S3 | During the nine months ended September 30, 2020, we granted approximately 0.3 restricted stock units that are contingent upon us achieving earnings targets over the three year period from 2020 to 2022 | 0.3 | grants in period | restricted stock units |
| S4 | Certain selling equity holders elected to receive deferred, variable earn out consideration with an estimated value of $21,500 over the rollover period of three years. | $21,500 | earn out consideration | estimated value |

Table 10: Instances of exact match and no match by the baseline approach. Column *Baseline* denotes the responses generated by the baseline approach.

47.77%, and 49.01% respectively vs. 27.59%).
We see that there is no significant improvement in performance as the number of parameters increases. T5 and Tk-Instruct Large (49.23% and 47.70%) give nearly the same F1 scores as BART (49.01%).

**Exact and No Match:** Table 9 gives the summarized results for exact and no match scores of all the approaches on EDGAR10-Q dataset. The results of ChatGPT and Baseline are consistently low, as shown in Table 16. We infer this is because of complex hidden contexts and the sentence structures of the dataset. ChatGPT's score is consistently lower than the baseline as the recall of ChatGPT is consistently higher due to which obtaining an exact match is difficult. Table 17 shows the results of supervised learning where consistently higher scores are obtained. Instruction-tuned variants of T5 (Flan and Tk-Instruct) perform the best out of the models but the overall score is still low.

**No Match:** The baseline results are consistently worse than ChatGPT, as shown in Table 18. The no-match score for the baseline is more than twice as compared to ChatGPT (47.96% vs. 19%). As shown in Table 19, the no-match score for supervised learning models is also around 20%. This could again be attributed to the recall scores, as all the supervised models and ChatGPT had recall scores of around 48.

Table 23 gives a few examples of both exact and no matches by the Baseline method.

## 6 ANALYSIS

In this section, we compare our approach with traditional OpenIE approaches and highlight the differences between them. We observe the effect of instruction tuning on the dataset and compare its performance. We explore the effects of the dataset with respect to different downstream tasks by using the models pre-finetuned on EDGAR on different downstream tasks.

### 6.1 CONTEXT-NER VS. OPENIE

Traditionally, information extraction approaches from textual documents assume pre-specified relations for a given domain and employ crowd-sourcing or distant supervision approaches Hoffmann et al. (2011); Liu et al. (2016) to collect examples and train models for each type of relation. However, these approaches have a limitation in that they cannot extract unseen relations that were not observed or specified during training, rendering them impractical. In contrast, Open Information Extraction (OpenIE) Etzioni et al. (2008) does not rely on pre-defined relations but extracts them on-the-fly as they are encountered. To compare our methods with existing OpenIE models, we evaluated Stanford's OpenIE and AllenNLP OpenIE models Stanovsky et al. (2018) on a subset of the EDGAR10-Q

| Dataset | Bloomberg GPT | T5 | EDGAR-T5 |
|---|---|---|---|
| FiQA SA | 75.07 | 74.89 | 80.42 |
| FPB | 51.07 | 55.77 | 79.69 |
| Headline | 82.20 | 90.55 | 93.55 |

Table 11: Comparison of EDGAR-T5 and Vanilla T5 on different finance related tasks. Both models are 770M is size. BloombergGPT 50B is used as the baseline. Scores are weighted F1 as shown in BloombergGPT 50B.

| Datasets | T5 Large | EDGAR T5 Large |
|---|---|---|
| Boolq | 32.94 | 37.41 |
| CB | 89.85 | 94.20 |
| COPA | 63.20 | 86.40 |
| RTE | 86.64 | 94.70 |
| WIC | 58.71 | 59.80 |

Table 12: Comparison of EDGAR-T5 and Vanilla T5 on downstream NLP tasks. Both models are 770M in size. Scores denote F1 score

dataset. Our findings indicate that Open IE models struggle when dealing with long-range dependencies. We applied both OpenIE frameworks to the sentences shown in Table 23 and present their results in Table 24. Notably, both frameworks failed to recognize any relations, contextual phrases, or entities.

## 6.2 MARGINAL IMPROVEMENT WITH INSTRUCTION TUNING

Following works from instruction tuning Wang et al. (2022b); Gupta et al. (2023), we add instructions on the train data and instruction tune Tk-Instruct. Figure 2 showcases the performance increase across the entire test set. Across each sentence category, there is an increase of roughly 2%, highlighting that instruction-tuned models with instruction data work well. The improvement is significant in sentences with 5+ categories where there is an increase of absolute 4%.

## 6.3 EFFECT OF EDGAR10-Q ON DOWNSTREAM TASKS

To study the impact of EDGAR10-Q in the real world, we compare the effect of a model pre-finetuned on EDGAR10-Q vs a vanilla T5 model. We call T5 pre-finetuned on the dataset as EDGAR-T5. Both EDGAR-T5 and vanilla T5 are then finetuned on three finance datasets; FiQASinha & Khandait (2021), FPBMalo et al. (2014), and Headline Maia et al. (2018) datasets. All the hyperparameters for vanilla T5 and Edgar-T5 were the same for a fair comparison[5]. We use BloombergGPT-50B Wu et al. (2023) 10 shot score as the baseline for these tasks. The splits used in downstream datasets and weighted F1 score were kept exactly the same as BloombergGPT for a fair comparison. As shown in Table 11, EDGAR-T5 outperforms both vanilla T5 and BloombergGPT on all three downstream tasks and establishes SOTA results on all three of the tasks.

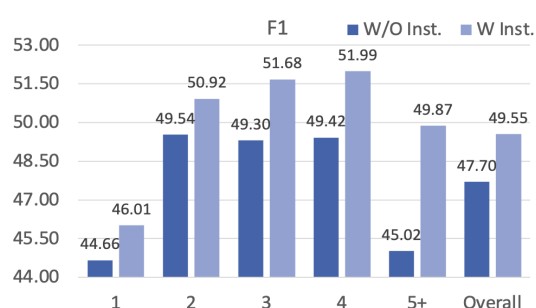

Figure 2: ROUGE-L F1 scores for showing the effect of instruction tuning Tk Instruct vs conventional finetuning. X-axis denotes scores on different numbers of entities and overall score.

The experiments suggest that the EDGAR10-Q dataset has led to an increase in the model's inherent ability for financial tasks. We also conducted a similar study to compare the performance of the models on general domain downstream NLP tasks. Table 12 shows the F1 score of the models across BoolqClark et al. (2019) , CBDe Marneffe et al. (2019), COPAGordon et al. (2012), RTEPilehvar & Camacho-Collados (2019) and WICPilehvar & Camacho-Collados (2019) datasets. Same hyperparameters were used for training of both the models. Performance gains were observed across all five datasets demonstrating the atifact's usability in general domain as well. We release all the finetuned models to the community for future use.

---

[5]Hyperparameters are available in appendix

## 6.4 Case Study : Human Evaluation

We also present a study of human evaluation on a small sample of the dataset (randomly sampled 100 examples). To account for human bias, we ask two graduate students evaluate the samples.

We get a Rouge-L F1 score of 34.69% (average) - indicating the difficulty of the task for non-experts. The exact match score is 8.1% while the no-match score is 36% - both further validating the task's difficulty.

## 7 Related Work

Several approaches have been developed for State-of-the-art NER detection Chawla et al. (2021); Li et al. (2019a); Luoma & Pyysalo (2020); Du et al. (2010); Zhu et al. (2018). Multiple approaches have been developed around various aspects of NER Moon et al. (2018); Amalvy et al. (2023); Li et al. (2020); Kocaman & Talby (2021); Zhong & Chen (2021); Zhang et al. (2022); Shon et al. (2022); Wang et al. (2022a). Etzioni et al. Etzioni et al. (2008) introduced a schemaless approach for extracting facts from text, focusing on relation extraction using OpenIE. However, this approach assumes relations between two entities, which poses challenges for financial data. Levy et al. Levy et al. (2017) used a zero-shot approach to train MRC model on templatized questions and inferenced it on unseen relations. Li et al. Li et al. (2019b) formalizes relation extraction as multi-turn question answering. Miwa et al. Miwa & Bansal (2016) jointly extracted entities and relations using neural networks, but performance suffers on unseen relations. Sun et al. Sun et al. (2018) build on the previously mentioned framework and uses a joint learning framework and a flexible global loss function to capture the interactions of the entities and their relationships. McCann et al. McCann et al. (2018) introduced decaNLP, addressing multiple challenges including relation extraction and question answering. Various frameworks like Stanford CoreNLP's NER, Spacy, NLTK, and Flair Manning et al. (2014); Honnibal et al. (2020); Loper & Bird (2002); Akbik et al. (2019) are available for entity extraction. We aim to extract entities and their contexts, a more complex scenario than relation-based approaches.

## 8 Conclusion, Limitations and Future Work

In this paper, we introduced the Context-NER task, which aims to bridge the gap between existing NER tasks by extracting relevant phrases for entities. We also presented the EDGAR10-Q dataset, which is a large and complex finance dataset, providing a valuable resource for research in this domain. Through our baseline approach, we demonstrated the feasibility of solving the Context-NER task and conducted extensive experiments to evaluate our method's performance. Our comparison with GPT-3 showcased the challenges posed by the dataset. Additionally, we explored a supervised setting by finetuning pre-trained language models, which showed promising results. We believe that the introduction of the EDGAR10-Q dataset and our study will encourage further investigation and advancements in this field.

To advance our work, there are several promising directions for future research. Due to limited computational resources, we were unable to finetune large models (greater than 1B) on the EDGAR10-Q dataset. Elaborate experimentation could be conducted using other instruction-based or chain-of-thought reasoning on the EDGAR10-Q dataset. Future work should consider leveraging more powerful models to potentially achieve higher scores on this dataset. Furthermore, our evaluation set for ChatGPT was smaller than the actual test set due to budget constraints. Lastly, expanding the dataset to include reports from different markets, non english languages and including more recent years would enable researchers to explore the generalizability and temporal dynamics of the task.

## Ethical Considerations

We have verified that all licenses of source documents used in this document allow their use, modification, and redistribution in a research context. There were no real-life names in the data set. No particular sociopolitical bias is emphasized or reduced specifically by our methods.

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

APPENDIX

## A  DATASET CREATION

The process to extract data is described below:

The code starts from the function called driver_writer_func, which takes five arguments: company_name, cik, start_date, base_folder, and dest_folder. The function performs the following steps:

1. Deletes the base_folder directory and creates a new one.

2. Creates a dest_folder directory if it does not already exist.

3. Calls a function get_all_submissions with arguments cik, start_date, base_folder, and company_name to retrieve financial documents for the company.

4. Parses the documents using IE_Parser and structures the resulting data in a tabular format.

5. Filters the data based on certain criteria such as text length and data type.

6. Parses the text data to extract entities, phrases, and questions and answers using various functions such as sent_parse, sentence_entity_flair, phrase_extraction, and qa_model (They are part of baseline extraction method and are explained in §).

7. Writes the resulting data to a CSV file in the dest_folder directory with the name company_name.csv.

**get_all_submissions:**  takes four arguments - cik (an integer), start_date, base_folder (a string), and company_name (also a string). It first checks if the company_name exists in a file called done_comps, and if so, prints a message saying all files of the company have already been downloaded and returns None. Next, it reads the contents of a file called DONE_LIST if it exists, and assigns it to the variable done_subs. Then, it converts cik to a string data type, calls the function get_accession_numbers with cik, '10-K', and start_date as arguments, and assigns the result to the variable subs_10k. Similarly, it calls get_accession_numbers with cik, '10-Q', and start_date as arguments and assigns the result to the variable subs_10q. It then concatenates these two lists (subs_10k and subs_10q) into a new list called subms. The function then logs the number of submissions made after start_date by cik. For each submission in the subms list, the function extracts the name and url of the JSON file associated with it. It then loads the JSON data into a dictionary called subm_json using the json.loads() method. From this dictionary, it extracts the list of files associated with the submission and filters out those with a .txt file extension. It then selects the first .txt file and extracts its file name and url. Next, it calls the function get_meta_data with the contents of the text file as an argument to extract metadata from the file. If successful, it assigns the submission type based on the extracted metadata. If the submission type folder doesn't exist in the base_folder, it creates the necessary directory structure. It then writes the contents of the text file to a file in the appropriate directory in the base_folder, and appends the name of the submission to a file called DONE_LIST. Finally, it appends the company_name to a file called DONE_COMP.

**get_accession_numbers:**  accepts three parameters: cik (a string), type (a string) and start_date (a datetime object). It returns a list of accession numbers for a company with the specified Central Index Key (CIK) that have been filed with the Securities and Exchange Commission (SEC) after a specified date and of a specified type. The function starts by constructing a URL based on the parameters passed in. The URL is used to fetch an HTML page containing a table of filing information. The function then processes the HTML page using BeautifulSoup to extract the relevant table, convert it to a pandas DataFrame, and filter the rows to those with filing dates greater than the specified start_date. It then extracts the accession numbers from the filtered table, cleaning them up and returning them as a list.

**get_meta_data:**  accepts a string subm_details_text that contains the content of a submission details text file in the EDGAR database. It returns a dictionary containing the metadata for the submission. The function starts by initializing an empty dictionary called meta_data and two lists called running_titles and running_indents. It then loops over the rows of the subm_details_text string,

splitting each row into segments using the colon (:) as a separator. If a row contains a single segment, the function assumes that it is a heading and adds it to running_titles along with its indentation level, which is calculated by counting the number of tabs in the row. If a row contains two segments, the function assumes that it is a key-value pair and adds it to meta_data using the deep_set function to create nested dictionaries for the various levels of headings. The deep_set function sets the value of a nested dictionary by walking the dictionary hierarchy according to the list of headings passed in and creating new dictionaries as needed. Finally, the function returns the meta_data dictionary.

## B   DATASET SCHEMA

### B.1   RAW DATA

```
{'paragraph': 'The Company has non trade receivables from certain of its manufacturing vendors resulting from the sale of compo
nents to these vendors who manufacture subassemblies or assemble final products for the Company. The Company purchases these co
mponents directly from suppliers. As of September 25, 2021, the Company had three vendors that individually represented 10% or
more of total vendor non trade receivables, which accounted for 52%, 11% and 11%. As of September 26, 2020, the Company had two
vendors that individually represented 10% or more of total vendor non trade receivables, which accounted for 57% and 11%.',
 'value': '52',
 'label': "['Concentration risk, percentage']",
 'name': 'us-gaap_ConcentrationRiskPercentage1',
 'type': 'percentItemType',
 'sent': 'As of September 25, 2021, the Company had three vendors that individually represented 10% or more of total vendor non
trade receivables, which accounted for 52%, 11% and 11%.',
 'entity': '52%',
 'entity_type_ext': 'PERCENT',
 'sentence': 'As of September 25, 2021, the Company had three vendors that individually represented 10% or more of total vendor
non trade receivables, which accounted for 52%, 11% and 11%.',
 'phrases': "['more % of total vendor trade receivables', 'more %', 'total vendor trade receivables']",
 'qa_temp': "['total vendor trade receivables', 0.8616288900375366, 'what is total vendor trade receivables ?', '52%, 11% and 1
1%']",
 'key': 'total vendor trade receivables',
 'score': 0.8616288900375366,
 'question': 'what is total vendor trade receivables ?',
 'answer': '52%, 11% and 11%'}
```

Figure 3: Illustration of the raw data in json format obtained after dataset collection.

Figure 3 shows one instance of the raw dataset. The complete dataset is present in the GitHub repository. Each column is described below:

1. paragraph: It contains the input string and the sentences surrounding it.

2. value: The numerical value of the entity whose context is going to be extracted from the sentence

3. label: A list of phrases, which describe the entity. In this case, the phrases are: 'Concentration risk, percentage'.

4. name:   A string representing the name of the value, in this case, is 'us-gaap_ConcentrationRiskPercentage1'.

5. type: Description of the data type of the value, in this case, is 'percentItemType'.

6. sent: The sentence that contains the entity.

7. entity: Entity extracted using NER library '52%'.

8. entity_type_ext: The data type of the entity extracted using the NER library, which is 'PERCENT'.

9. sentence: Cleaned version of sent.

10. phrases: Phrases extracted from the phrase generation algorithm, including 'more % of total vendor trade receivables', 'more %', and 'total vendor trade receivables'.

11. qa_temp: List of questions that are formed using the phrases.

12. key: The phrase whose question gave the correct answer.

13. score: The confidence score given by the answer, that is 0.8616288900375366 in this case.

---

**Algorithm 2:** Phrase Generation *Pseudocode*

---

**Input:** Sentence
**Output:** List of Phrases

```
1  Function simple_noun_phrase_extractor(Sentence):
2      doc = sequence_of_token(Sentence),phrase_list = []
3      for token in Doc:
4          phrase = '  '
5          if token.head.pos in [Noun, Pronoun] and token.dep in [Object, Subject]:
6              for subtoken in token.children:
7                  if subtoken.pos is Adj or subtoken.dep is Comp:  phrase += subtoken.text + '  '
8              if len(phrase) is not 0:  phrase += token.text
9          if len(phrase) is not 0 and phrase doesnot have entities:  phrase_list.append(phrase)
10     return phrase_list
11 Function complex_noun_phrase_extractor(Sentence):
12     doc = sequence_of_token(Sentence)
13     phrase_list = []
14     for token in Doc:
15         if token.pos is Preposition:
16             phrase = '  '
17             if token.head.pos in [Noun, Pronoun]:
18                 for subtoken in token.head.children:
19                     if subtoken.pos is Adj or subtoken.dep is Comp:
20                         phrase += subtoken.text + '  '
21                 phrase += token.head.text + '  '+ token.text
22                 for right_tok in token.rights:
23                     if right_tok in [Noun, Pronoun]:
24                         for subtoken in right_tok.children:
25                             if subtoken.pos is Adj or subtoken.dep is Comp:
26                                 phrase += subtoken.text + '  '
27                         phrase += '  '+ right_tok.text
28                 if len(phrase) is > 1 and phrase doesnot have entities:
                       phrase_list.append(phrase)
29     return phrase_list
```

---

14. question: the question that gave the correct answer, that is, 'What is total vendor trade receivables ?' in this case.

15. answer: String that represents the MRC output of the BERT model used in the baseline approach.

## B.2 SUPERVISED MODELING DATA

Supervised modeling data consisted of the entity concatenated with the sentence. The output is one of phrases from the labels. In this case, the input is: *52%.As of September 25, 2021, the Company had three vendors that individually represented 108 or more of total vendor non trade receivables, which accounted for 52%, 11%, and 11%.* The output for this sentence is: *Concentration risk*

**Supervised Training setup:** We finetune generative models (T5 Base, T5 Large (Raffel et al., 2020), BART Base (Lewis et al., 2020), Flan-T5 Large (Chung et al., 2022), Tk-Instruct Large (Wang et al., 2022b)) on the train split of the dataset.

**Hyper parameters**: Train Batch Size: 8, Gradient Accumulation Steps: 8, Max Source Length: 512, Max Target Length: 128, Number of Epochs: 2, Warmup Steps: 100, Learning Rate: $\{5\}e-5$

## C BASELINE APPROACH

Consider the sentences:

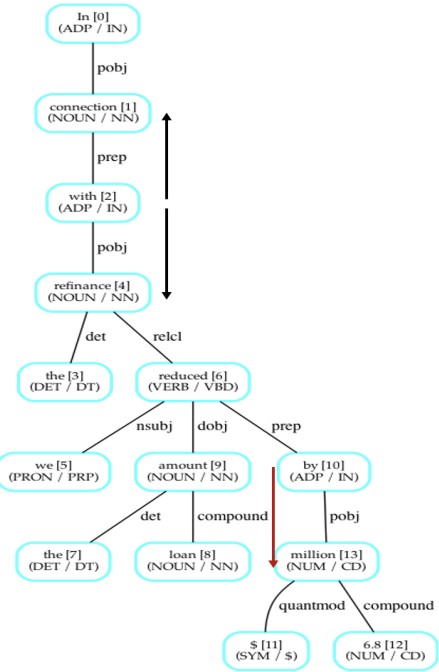

Figure 4: Syntactic tree structure for extraction of simple and complex noun phrases.

- In October 2019, the Company increased the borrowing capacity on the revolving credit loan by $33,000 increasing the available credit facility from $60,000 to $93,000.
- If the loan is paid during months 13-24 or 25-36 and then a penalty of 2% and 1%, respectively, of the loan balance will be charged on the date of repayment.
- The weighted-average remaining lease term and discount rate related to the Company's lease liabilities as of September 26, 2020 were 10.3 years and 2.0%, respectively.

## C.1 PHRASE GENERATION

This paper presents a simple, yet efficient technique to extract entities and their descriptions from sentences. As shown in Figure 1, it starts with data cleaning and entity extractions. A noun phrase (NP) (Stuart et al., 2013) includes a noun, a person, place, or thing, and the modifier that distinguishes it. Open IE is predicated on the idea that the relation (which is action verbs in most cases) is the central element of the study, from which all other considerations flow. However, in many cases, the verb is not helpful, particularly in financial data. Consider the sentence: "Deferred revenue for 2020 is $20 billion." Like most financial records are of the form "is, was, be," etc., the verb "is" in this sentence is an auxiliary verb and does not describe any particular event or give any information about the entity.

We extract two types of phrases from the sentences, namely simple and complex. In simple phrase extraction, each sentence comprises subject-object and verb connecting them where Subject or Object is usually a noun or pronoun. After searching for a noun and pronoun, we check for any noun compound or adjective. On the other hand, for complex phrase extraction we first start with preposition extraction. We then follow similar steps as in simple phrase extraction to look for phrases in both left and right of the preposition. It has to be noted that simple phrases are not always found on both sides of the proposition. Algorithm 2 further summarizes the process of simple and complex phrase extraction from the sentences.

Now we demonstrate the extraction of simple and complex noun phrases for the sentence, *'In connection with the refinance we reduced the loan amount by $6.8 million.'*. The syntactic tree for the above sentence is shown in Figure 4. We search if the token's POS tag is a noun or pronoun as we are looking just for noun phrases. We also ensure that phrase lies either in the Subject or Object of

the sentence to ensure we are skipping the relations. In this case, we got *"amount"* the first word of the phrase. After that, we iterate the node to see its children named subtoken in Algorithm 2. We search for subtoken's dependency relation with the token as a compound relation, or we search if the subtoken is an adjective. The intuition behind this is that if the subtoken and token have a compound relationship, they form a meaningful noun phrase. In this case, "amount" has a compound relationship with its subtoken *"loan"* so they together form *"loan amount"* as the meaningful noun phrase. Similar logic is followed for searching adjectives. Complex NPs are identified as series of noun phrases with a preposition separating them, so we start by identifying them. In this example, the preposition identified was *"in"*. Then we iterate both up and down the node to find noun phrases that follow the same method mentioned above. The noun phrases identified from the top were *"connection"* and the bottom was *"refinance"*. The entire complex NP was formed as NP from top + preposition in the middle and + NP from below. The resultant was *"connection with refinance"*.

## C.2 Machine Reading Comprehension Model

This paper presents a zero-shot technique as we leverage the phrase generation to generate meaningful questions without further training of the machine reading comprehension (MRC) model. This allows our technique to be domain agnostic and thus can be easily expanded to newer domains. The process to leverage noun phrases to generate the questions and further using the MRC model to associate entities with their corresponding descriptions is described below:

- Each paragraph in the document is broken down into sentences. For each sentence, the following are extracted: Phrases (using simple and complex noun phrases described in Algorithm 2) and Entities using the Flair NER Model.
- On the basis of the entity type and the noun phrases, the questions are framed accordingly. For instance, if the entity found out was of type date, then the question would be "when is" + NP?. In our example, the question for the first sentence for §C would be "how much is borrowing capacity on revolving credit loan ?".
- In instances where the entity type is of integer, float, or percent where appending "when is" or "how much is" does not give an advantage. For such cases, to keep the question generic we append "what is" to the noun phrase. For example, the question for second sentence for §C is, "What is the loan balance?" was created based on the entity type of 2% and 1%.
- Once these questions are generated, they are fed into the MRC Model, and its answer is checked if it contains the entity. To give an example, in the 1st sentence, the following questions are created, and the model returns their corresponding answers and their confidence values:
  - "How much is borrowing capacity on revolving credit loan?" answer: "$33,000", confidence score: 0.946
  - "How much is borrowing capacity ?" answer: "$33,000", confidence score: 0.824
  - "How much is revolving credit loan ?" answer: "$33,000", confidence score: 0.856
  - "How much is available credit facility ?" answer: "$60,000 to $93,000", confidence score: 0.5762

  If there are multiple questions whose answer has the entity, we select the question whose answer is of the highest confidence value. In the above example, "borrowing capacity on revolving credit loan" is chosen as the key for $30,000, and "revolving credit loan" is chosen as the key for both $60,000 to $93,000.
- If the entity is not present in the response of the MRC model, the question is discarded. In the 2nd Sentence of Table 1, the following questions are created :
  - "What is penalty of % ?"
  - "What is loan balance ?"

  None of them are returning "13-24 or 25-36", so the phrases "penalty of %" and "loan balance" are discarded.
- There are instances where none of the generated questions returned an answer with the target entity or returned responses with a different entity as shown above. For those cases, we create the question "what is" entity?. Here, its response would be considered as the key

| Phrases extracted | Question | Answer |
|---|---|---|
| borrowing capacity, available credit facility | What is borrowing capacity on evolving credit loan ? | $60,000 to $93,000 |
| borrowing capacity on revolving credit loan | How much is available credit facility ? | $33,000 |
| penalty of %, loan balance date of repayment | What is 13-24 or 25-36 ? What is 2% and 1% | loan is paid during months 2% (Wrong Answer) |
| lease liabilities, discount rate average lease term | What is average lease term ? What is discount rate ? | 10.3 years 2.00% |

Table 13: Illustration of the baseline approach based on sentences in §C

(opposite to the case above). In the 2nd sentence of the Table, none of the questions returned relevant answers, So the following questions were created:

– "What is 13-24 or 25-36 ?"
– "What is 2% and 1% ?"

- In the above cases, where questions are formed based on entities, the answers are checked if they have given any other entity as the answer. For instance, the questions, "what is 2% and 1% ?" return "2" as the answer to the second sentence. If the cases mentioned above hold, then the response is discarded. Here all the cases to identify the noun phrase associated with the entity fail, so no answer is returned.

- If they do not fail, then the response is also considered a viable answer. For instance, In the 2nd sentence, the question was framed: "What is 13-24 or 25-36 ?" which returned "loan is paid during months" as the answer.

Using the rules stated above, the entity and its associated noun phrases are identified. The last two columns of Table 13 show the questions which were generated and their responses from the MRC model. Inspired by the success of the pre-trained transformer model, we employ distilled BERT (Sanh et al., 2019) by Hugging Face (Wolf et al., 2020) trained on SQuAD dataset (Rajpurkar et al., 2016) as the MRC model for our zero-shot question answering [6].

## C.3 Baseline Method Code:

**sent_parse:** This function takes a row as input, which has a column named "paragraph", and tokenizes the paragraph into sentences using the sent_tokenize function from the nltk library. It then iterates through each sentence and checks if the value in the row is present in the sentence. If it is, the function returns the sentence. If not, the function does nothing.

**sentence_entity_flair(sentence,entity, entity_type):** This function takes a sentence, an entity, and an entity type as input. It first removes words between parentheses that do not contain digits, as well as any forward slashes. It then removes any brackets surrounding a dollar amount. The function then uses an entity_tagger function to identify entities in the sentence, and iterates through each identified entity. Depending on the entity label and the entity type provided, the function checks if the entity matches the given entity. If it does, the function creates a list containing the entity, its label, and the original sentence, and returns it. If no matching entity is found, the function returns a list containing the original entity, a label of "none", and the original sentence.

**preposition_phrase_extraction:** This function takes a text as input and uses the nlp function from the spacy library to parse the text. It then iterates through each token in the parsed text, and if the token is an adposition (preposition), it checks if its headword is a noun or pronoun. If it is, the function creates a phrase by appending any adjectives or compound dependencies of the head noun, the head noun itself, and any nouns or proper nouns to the right of the preposition, along with the preposition. The function then returns a list of all phrases found.

---

[6] Hugging Face's Model Link: https://huggingface.co/transformers/v2.8.0/usage.html.

**noun_phrase_extraction:** This function takes a text as input and uses the nlp function from the spacy library to parse the text. It then iterates through each token in the parsed text, and if the token is a noun or proper noun and its dependency is either "dobj," "pobj," "nsubj," or "nsubjpass," the function creates a phrase by appending any adjectives or compound dependencies of the noun and the noun itself. The function then returns a list of all phrases found.

**phrase_extraction:** This function takes a text as input and uses the entity_tagger function to identify entities in the text. It then uses the preposition_phrase_extraction and noun_phrase_extraction functions to extract phrases from the text. For each extracted phrase, the function checks if it is present in any of the identified entities. If it is not, the function appends the phrase to a list of phrases to return. The function then returns the list of phrases.

# D PROMPTS AND INSTRUCTIONS

## D.1 INSTRUCTIONS FOR HUMAN EVALUATIONS

> **Instructions to Human Annotators**
> You are given an entity and a sentence that contains the entity. You job is to generate a phrase that describes the meaning of the entity in the sentence. The phrase may not be present in the sentence and you may have to come up with phrase using your prior knowledge. An example is given to help you out:
> **Example: Input:** $15. Issuance of common stock in May 2019 public offering at $243.00 per share, net issuance costs of $15.
> **Output:** Common stock public offering issuance cost

## D.2 CHATGPT INSTRUCTIONS

> **Zero shot**
> **Definition:** You are given a "key term" and a sentence. Based on the information in the sentence, output a brief description of the role of the "key term" in the context of the sentence. The output should be a brief relevant phrase describing the "key term" within a given sentence, regardless of whether the phrase is explicitly present in the sentence.
>
> **One shot**
> **Example: Input:** $15. Issuance of common stock in May 2019 public offering at $243.00 per share, net issuance costs of $15.
> **Output:** Common stock public offering issuance cost
>
> **Few shot**
> **Example 1:**
> **Input:** $15. Issuance of common stock in May 2019 public offering at $243.00 per share, net issuance costs of $15.
> **Output:** Common stock public offering issuance cost
> **Example 2:**
> **Input:** $1.8 billion. As of December 31, 2021 and 2020, the net carrying value of real estate collateralizing our mortgages payable totaled $1.8 billion.
> **Output:** Net carrying value of real estate collateralizing the mortgages payable
> **Example 3:**
> **Input:** 75,305,400. $0.00001 par value— 76,420,805 and zero shares authorized as of December 31, 2020 and September 30, 2021, respectively; 75,305,400 and zero shares issued and outstanding as of December 31, 2020 and September 30, 2021, respectively; and aggregate liquidation preference, $464,036 and zero as of December 31, 2020 and September 30, 2021, respectively
> **Output:** shares outstanding

Figure 5 shows the precision and recall scores of instruction tuning Tk-Instruct on EDGAR10-Q. Similar trends are observed for precision and recall as for F1 score.

| # of | Baseline Scores | | |
|---|---|---|---|
| Ent. | P | R | F1 |
| 1 | 36.69 | 27.04 | 28.19 |
| 2 | 36.57 | 29.53 | 29.66 |
| 3 | 32.67 | 25.97 | 26.48 |
| 4 | 29.59 | 24.11 | 24.27 |
| 5+ | 23.82 | 19.35 | 19.56 |
| Overall | 34.59 | 27.06 | 27.59 |

Table 14: Score of Baseline approach on the test set showing precision, recall and F1

| # of | # of | Avg. S. | Chat GPT Scores | | |
|---|---|---|---|---|---|
| Ent. | Sent. | Len. | P | R | F1 |
| 1 | 38919 | 29.68 | 25.13 | 42.19 | 28.55 |
| 2 | 24717 | 33.61 | 26.25 | 50.99 | 31.53 |
| 3 | 8131 | 39.03 | 26.60 | 48.86 | 31.54 |
| 4 | 3341 | 50.28 | 26.56 | 50.37 | 31.62 |
| 5+ | 1050 | 75.99 | 20.47 | 43.53 | 25.02 |
| Overall | 76158 | 33.49 | 25.72 | 47.49 | 30.31 |

Table 15: Details of ChatGPT performance on a smaller test set. The table shows the smaller test's statistics and ChatGPT's precision, recall, and F1.

| # of Entities | Baseline | ChatGPT |
|---|---|---|
| 1 | 4.88 | 0.84 |
| 2 | 6.58 | 1.28 |
| 3 | 5.97 | 1.29 |
| 4 | 5.44 | 1.84 |
| 5+ | 5.46 | 0.81 |
| Overall | 5.77 | 1.18 |

Table 16: Exact match of baseline approach and ChatGPT.

| # of Entities | Bart Base | T5 Base | T5 Large | Flan T5 Base | Tk Inst Base | Tk Inst w. Inst |
|---|---|---|---|---|---|---|
| 1 | 17.46 | 15.82 | 17.03 | 16.05 | 16.02 | 17.35 |
| 2 | 23.31 | 22.12 | 23.52 | 22.56 | 22.23 | 24.04 |
| 3 | 21.40 | 19.29 | 21.42 | 19.97 | 19.66 | 22.20 |
| 4 | 21.76 | 20.80 | 23.39 | 20.74 | 20.66 | 23.61 |
| 5+ | 21.43 | 18.38 | 20.42 | 18.21 | 17.90 | 22.09 |
| Overall | 20.88 | 19.31 | 20.92 | 19.64 | 19.44 | 21.45 |

Table 17: Exact match scores of supervised learning models. Tk Inst w. Inst denotes instruction tuning TkInstruct.

| # of Entities | Baseline | ChatGPT |
|---|---|---|
| 1 | 43.56 | 20.64 |
| 2 | 45.40 | 17.76 |
| 3 | 51.95 | 17.42 |
| 4 | 55.94 | 18.54 |
| 5+ | 64.26 | 24.99 |
| Overall | 47.96 | 19.0 |

Table 18: No match of baseline approach and ChatGPT.

| # of Entities | Bart Base | T5 Base | T5 Large | Flan T5 Base | Tk Inst Base | Tk Inst w. Inst |
|---|---|---|---|---|---|---|
| 1 | 25.44 | 25.46 | 24.65 | 25.31 | 25.44 | 24.44 |
| 2 | 23.39 | 23.68 | 22.73 | 23.60 | 23.62 | 22.91 |
| 3 | 21.39 | 22.17 | 21.29 | 22.50 | 22.53 | 20.17 |
| 4 | 23.18 | 23.15 | 21.94 | 23.40 | 23.05 | 21.51 |
| 5+ | 24.88 | 26.74 | 27.07 | 29.51 | 28.99 | 23.83 |
| Overall | 23.74 | 24.09 | 23.24 | 24.24 | 24.23 | 22.83 |

Table 19: No match scores of supervised learning models. Tk Inst w. Inst denotes instruction tuning TkInstruct.

# E    DETAILED RESULTS

## E.1    DETAILED BASELINE RESULTS

Table 14 give the detailed baseline results with entity wise scores. Precision uniformly decreases from 36% to 23% while recall is ranging from 19% to 30%, leading to an overall F1 score in the range from 20% to 30% for each category of the baseline model. F1 shows a linearly decreasing trend with an increase in the number of entities (the exception being of 2 entity sentences higher than 1).

Table 15 shows ChatGPT scores which is evaluated on a subset of the actual Test set. This evaluation set is roughly 75% of the actual test and is reduced due to budget limitations. However, the distribution of this eval set is similar to the actual set. Contrasting to the baseline approach, ChatGPT's F1 score stays constant at 31% and decreases sharply as the number of entities increases to 5+. The overall precision of ChatGPT is lower than baseline, but recall is much higher than baseline, resulting in a higher overall F1. The performance difference between the two increases as the length of sentences increases. Although the F1 of ChatGPT is higher than the baseline score (30.31% vs. 27.59%), there is significant room for improvement.

## E.2    FINE TUNING RESULTS

Figure 5 shows the precision and recall results of further instruction tuning.

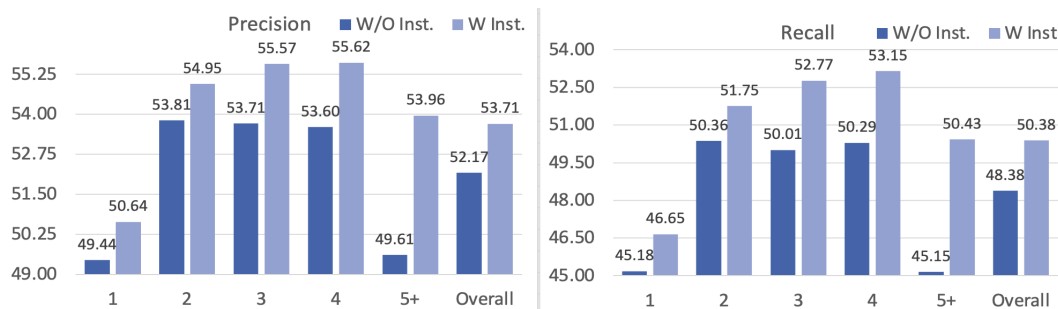

Figure 5: ROUGE-L precision recall scores for showing the effect of instruction tuning Tk Instruct vs conventional finetuning. X-axis denotes scores on different numbers of entities and overall score.

| Bigrams | | Trigrams | | Four-grams | |
|---|---|---|---|---|---|
| Phrase | Count | Phrase | Count | Phrase | Count |
| ('december', '31') | 485,450 | ('december', '31', '2020') | 155,552 | ('months', 'ended', 'june', '30') | 114,326 |
| ('months', 'ended') | 315,352 | ('december', '31', '2019') | 131,680 | ('months', 'ended', 'september', '30') | 90,217 |
| ('june', '30') | 267,539 | ('december', '31', '2021') | 131,652 | ('six', 'months', 'ended', 'june') | 84,985 |
| ('31', '2020') | 222,873 | ('ended', 'june', '30') | 124,808 | ('nine', 'months', 'ended', 'september') | 68,952 |
| ('september', '30') | 209,254 | ('months', 'ended', 'june') | 119,594 | ('years', 'ended', 'december', '31') | 57,929 |
| ('31', '2021') | 203,014 | ('three', 'months', 'ended') | 118,595 | ('months', 'ended', 'march', '31') | 50,183 |
| ('march', '31') | 171,484 | ('ended', 'december', '31') | 109,878 | ('three', 'months', 'ended', 'march') | 46,313 |
| ('31', '2019') | 158,276 | ('six', 'months', 'ended') | 104,118 | ('three', 'six', 'months', 'ended') | 46,075 |
| ('common', 'stock') | 155,975 | ('ended', 'september', '30') | 98,364 | ('2021', 'december', '31', '2020') | 45,670 |
| ('30', '2021') | 151,148 | ('months', 'ended', 'september') | 95,356 | ('december', '31', '2021', '2020') | 43,931 |
| ('30', '2020') | 141,908 | ('nine', 'months', 'ended') | 89,316 | ('year', 'ended', 'december', '31') | 39,470 |
| ('ended', 'june') | 132,265 | ('june', '30', '2021') | 71,256 | ('december', '31', '2020', '2019') | 38,917 |
| ('2020', '2019') | 130,201 | ('september', '30', '2020') | 68,991 | ('three', 'nine', 'months', 'ended') | 38,805 |
| ('million', 'the') | 130,173 | ('september', '30', '2021') | 68,391 | ('2020', 'december', '31', '2019') | 35,902 |
| ('million', 'as') | 129,389 | ('million', 'the', 'company') | 66,153 | ('december', '31', '2019', '2018') | 33,365 |
| ('three', 'months') | 125,415 | ('june', '30', '2020') | 65,569 | ('ended', 'september', '30', '2020') | 33,168 |
| ('30', '2019') | 120,446 | ('june', '30', '2022') | 62,811 | ('million', 'as', 'december', '31') | 32,953 |
| ('the', 'company') | 119,993 | ('as', 'december', '31') | 59,171 | ('ended', 'december', '31', '2020') | 32,933 |
| ('million', 'respectively') | 115,122 | ('september', '30', '2019') | 59,167 | ('million', 'three', 'months', 'ended') | 32,800 |
| ('ended', 'december') | 113,831 | ('years', 'ended', 'december') | 58,640 | ('ended', 'june', '30', '2021') | 32,246 |

Table 20: We present some additional descriptive statistics about the EDGAR-10Q dataset here. Columns 1,2 and 3 present a list of top 20 most common bi-grams,tri-grams and 4-grams respectively (along with their counts).

# F  ADDITIONAL ANALYSIS OF EDGAR10-Q

Table 20 provides list of top 20 bigrams, trigrams and 4 grams of the datasets, along with their count.

# G  CHATGPT RESPONSE FOR EDGAR10-Q IN QA FORMAT

In this section we conduct a case study on format conversion where we find that ChatGPT fails to do this task on converting this to a QA task ("What is entity?" keeping sentences as the context).

Based on the sentence given below, answer the following question:

**Sentence:** During the nine months ended September 30, 2020, we granted approximately 0.3 restricted stock units that are contingent upon us achieving earnings targets over the three year period from 2020 to 2022.
**Question:** What is 0.3 ?

**Response:** In the given sentence, "0.3" refers to the number of restricted stock units that were granted during the nine months ended September 30, 2020.
_____________________________________________________________________________________

**Sentence:** Certain selling equity holders elected to receive deferred, variable earn out consideration with an estimated value of $21,500 over the rollover period of three years.
**Question:** What is 21,500 ?

**Response:** The amount $21,500 represents the estimated value of deferred, variable earn out consideration that certain selling equity holders elected to receive. This amount will be distributed over a period of three years during the rollover period.

## H  ANNOTATION PROCESS OF THE DATASET

The gold labels in our dataset are meticulously created by human annotators who are domain experts and possess a comprehensive understanding of the organizations involved. The annotation process follows guidelines from the SEC (U.S. Securities and Exchange Commission) for filing reports. As highlighted on the SEC website https://www.sec.gov/oiea/investor-alerts-and-bulletins/how-read-10-k10-q?_gl=1*32qw6m*_gcl_au*MTc5ODMzNDg2Ni4xNjk5MDYzOTU1 , "U.S. companies are required to present their financial statements according to a set of accounting standards, conventions and rules known as Generally Accepted Accounting Principles, or GAAP. An independent accountant audits the company's financial statements. For large companies, the independent accountant also reports on a company's internal controls over financial reporting."

Importantly, the gold labels are not automatically extracted from the annual reports; rather, they are carefully annotated by human experts following the guidelines and procedures outlined in the SEC regulations. Given the critical nature of these reports in depicting an organization's financial well-being, they are prepared with utmost care, often under the oversight of the management team. As emphasized on the same SEC website,"Laws and regulations prohibit companies from making materially false or misleading statements. Likewise, companies are prohibited from omitting material information that is needed to make the disclosure not misleading. In addition, a company's CFO and CEO must certify to the accuracy of the 10-K and 10-Q."

To ensure consistency, all instances in the dataset adhere to the guidelines provided by the SEC filing section for public organizations. These guidelines are outlined in the "Prepare filing documents" section, and we have included a reference to this information in our revised paper. For convenience and reference, the specific SEC guidelines can be found at https://www.sec.gov/edgar/filer-information/how-do-i

## I  ADDITIONAL EXPERIMENTS

### I.1  EFFECT OF CONTINUAL PRETRAINING

We did continual pre training with 100K samples of the dataset (we call this mode Cont. PT EDGAR T5) and then later fine tuned with the same downstream financial tasks described in the paper. The results presented in Table 21 were better than vanilla T5 but lesser than EDGAR T5.

### I.2  BASELINE WITH FINBERT

In this experiment we use FinBERT instead of vanilla BERT for our experimentation. The results are given in Table 22 :

| Dataset | Cont. PT EDGAR T5 | T5 | EDGAR-T5 |
|---|---|---|---|
| FiQA SA | 75.99 | 74.89 | 80.42 |
| FPB | 57.47 | 55.77 | 79.69 |
| Headline | 91.99 | 90.55 | 93.55 |

Table 21: Comparison of EDGAR-T5 and Vanilla T5 and Continual Pretrained EDGAR T5

| # of Ent. | FinBERT Baseline Scores | | |
|---|---|---|---|
| | P | R | F1 |
| 1 | 36.59 | 27.12 | 28.21 |
| 2 | 36.34 | 29.56 | 29.90 |
| 3 | 32.54 | 25.92 | 26.49 |
| 4 | 29.32 | 24.62 | 24.53 |
| 5+ | 23.89 | 19.98 | 19.82 |
| Overall | 35.89 | 26.56 | 27.28 |

Table 22: Score of FinBERT Baseline approach on the test set showing precision, recall and F1

| | Sentence | Entity | Labels | EDGAR T5 |
|---|---|---|---|---|
| | **Instances of Exact Match** | | | |
| S1 | Premium receivables are reported net of an allowance for doubtful accounts of $250 and $237 at September 30, 2020 and December 31, 2019, respectively. | $250 and $237 | premium receivable | premium receivable |
| S2 | The fair value of the collateral received at the time of the transactions amounted to $1,019 and $351 at September 30, 2020 and December 31, 2019, respectively. | $1,019 and $351 | fair value of collateral | fair value of collateral |
| | **Instances of No Match** | | | |
| S3 | During the nine months ended September 30, 2020, we granted approximately 0.3 restricted stock units that are contingent upon us achieving earnings targets over the three year period from 2020 to 2022 | 0.3 | grants in period | stock units |
| S4 | Certain selling equity holders elected to receive deferred, variable earn out consideration with an estimated value of $21,500 over the rollover period of three years. | $21,500 | earn out consideration | deferred consideration |

Table 23: Instances of exact match and no match by the baseline approach. Column *Baseline* denotes the responses generated by the baseline approach.

### I.3 CASE STUDY OF MODEL MAKING ERRORS

### I.4 OPEN IE RESULTS

### I.5 REGARDING SUPERVISED FINETUNING

The fine-tuning is conducted through supervised finetuning. In this process, both the entity and the sentence containing the entity are selected as prompts, and the corresponding phrase describing the entity is chosen as the response.

| Subject | Relation | Object |
|---|---|---|
| Stanford OpenIE Angeli et al. (2015) | | |
| Premium receivables | are reported | net of allowance |
| fair value | received at | time of transactions |
| we | granted | approximately 0.3 stock units |
| variable | earn out | consideration |
| Allen AI OpenIE Stanovsky et al. (2018) | | |
| Not Found | are | Not Found |
| the collateral | received at | at the time of the transactions |
| Not Found | restricted | stock units |
| Not Found | estimated | value |

Table 24: Responses of different OpenIE approaches on EDGAR 10-Q examples in Table 24.

