# OpenReview forum: "ContextNER: Contextual Phrase Generation at Scale"
_ICLR.cc/2024/Conference — Submitted to ICLR 2024_

### Official Review · Reviewer_btb9 · 2023-10-30

**Soundness:** 2 fair
**Presentation:** 2 fair
**Contribution:** 2 fair
**Rating:** 3
**Confidence:** 2

**Summary:**

The paper proposes a new task in which entities mentioned in text need to be described. This is helpful for non-experts to understand quantities and dates in financial text. The authors introduced new dataset called EDGAR10-Q of sentences extracted from annual and quarterly financial reports. The entities were automatically extracted and described by experts. Besides, the authors introduced a baseline based on machine reading comprehension model, which doesn't require to train. The authors carried out experiments with several systems: the proposed baseline, few-shot chatgpt, some generative models (e.g. T5). The authors performed some analyses and found that pre-finetuning on EDGAR10-Q (e.g. using T5) is beneficial for some financial downstream tasks, even better than using LLM BloombergGPT 50B.

**Strengths:**

The paper is original in the sense that the proposed task is new.

**Weaknesses:**

One of the main limitation of the paper is its dataset, EDGAR10-Q.

* First of all, it is unclear how the data were annotated. The paper emphasizes that "These reports are prepared by domain experts (financial analysts), ensuring highest quality of gold labels and contextual relevance." but nowhere in the paper mentions the job of experts. The paper also does not presents the quality of annotation.

* Secondly, it is not clear why entity descriptions can contain information outside of the given sentence. For instance, in "There were impairments of \\$0.8 million for the three months ended June 30, 2020 and [\\$2.2 million] for the six months ended June 30, 2020.", why could "\\$2.2 million" can be "valuation allowance for loan"? What is the evidence for that? If the reason is of domain "common sense", how many experts agree with that annotation? The major concern is that if the sentence itself doesn't contain any evidence, a model trained on this dataset can generate hallucination.


Another limitation is that it is unclear about why the baseline is introduced. In the experiments, the baseline is clearly outperformed. The baseline doesn't have any contribution to the community either.

Writing is also a major limitation. The paper is difficult to follow as its main text is not self-sufficient. To understand the main text, readers need to check appendices. For instance, the main text doesn't introduce the used MRC model, how BERT is used in that model. In the experiments, how were the used generative models finetuned? The number 30.31% (last paragraph page 6) isn't introduced at all in the main text. In additions, there are several typos such as "Additionally, In..." (sec 2).

**Questions:**

In section 6.3., the authors "[...] use BloombergGPT-50B Wu et al. (2023) 10 shot score as the baseline". What does it mean by "10 shot score"?

How was T5 finetuned on  EDGAR10-Q? Is that supervised finetuning?

---

> ### Author Response · Authors · 2023-11-23
> **Response to Reviewer btb9**
>
> **R3.1: "Dataset Annotation Process"** Please check general response G1.1
>
> **R3.2: "Concerns about Baseline"** Inspired by various benchmark papers (example DROP paper [1]), we wanted to have a holistic evaluation of the task and the dataset. This approach allows for a comprehensive evaluation and comparison of different models, providing a benchmark for assessing the performance of proposed methodologies. We have followed this convention to ensure a thorough understanding of the landscape and to demonstrate the effectiveness of our proposed models.
> Additionally, the introduction of the baseline serves the purpose of exploring a low-resource setting, where achieving scores comparable to state-of-the-art models like ChatGPT is valuable without requiring an extensive budget. This aspect is particularly relevant for scenarios with resource constraints, and the baseline contributes to the versatility and applicability of our proposed approach.
> We appreciate your thoughtful consideration, and if you have any further questions or suggestions, please feel free to share them.
>
> 1. https://aclanthology.org/N19-1246.pdf
>
> **R3.3: "Regarding issues in writing and typos"** We have carefully addressed the typos you pointed out in the revised draft and are actively collaborating with native speakers to further enhance the overall writing quality.
> To specifically address your concern about the paper's difficulty to follow, we acknowledge the importance of self-sufficiency in the main text. We understand that key elements, such as the introduction of the used Machine Reading Comprehension (MRC) model and the fine-tuning process of generative models, were not adequately covered in the main text. We have revised the manuscript to include a more comprehensive explanation of these aspects, making the main text more self-sufficient.
> Furthermore, we have taken note of your observation regarding the number 30.31%, which was not introduced in the main text. We have rectified this issue by incorporating an explanation of this value in the relevant section of the main text for improved clarity.
> We welcome any additional suggestions or feedback you may have to further enhance its readability.
>
> **R3.4: "Bloomberg GPT 10 shot score"** Thank you for your inquiry. The term refers to the process of selecting 10 instances along with their corresponding gold labels from the training set. These instances are then used as in-context examples during the inference phase.
> For further clarification and context, we recommend referring to the Bloomberg GPT paper by Wu et al. (2023) [2], which provides additional details on the methodology and procedures used, including the specific approach to scoring with a 10-shot setup.
> We appreciate your attention to this aspect of our work, and if you have any more questions or require further clarification, please feel free to reach out. We are committed to ensuring the comprehensibility of our paper and welcome any feedback or inquiries.
> 2. https://arxiv.org/abs/2303.17564
>
> **R3.5: "Finetuning of T5 on EDGAR10-Q"**
> Your understanding is accurate; the fine-tuning is conducted through supervised finetuning. In this process, both the entity and the sentence containing the entity are selected as prompts, and the corresponding phrase describing the entity is chosen as the response.
> We have explicitly detailed this process in the revised draft of our paper. If you have any further questions or if there are additional points you would like us to elaborate on, please feel free to let us know.

---

### Official Review · Reviewer_uqH6 · 2023-11-01

**Soundness:** 2 fair
**Presentation:** 3 good
**Contribution:** 2 fair
**Rating:** 3
**Confidence:** 3

**Summary:**

This paper introduces a new task that aims to generate the relevant context for entities in a sentence and builds a dataset called EDGAR10-Q, which contains 1M sentences, 2.8M entities, and an average of 35 tokens per sentence sourced from the finance domain. The authors conducted comprehensive experiments and proved that the EDGAR10-Q dataset is challenging for LLMs. Furthermore, they also built a baseline for the dataset.

**Strengths:**

- The paper is well-organized and easy to follow.
- Experiments seem to be solid and comprehensive.

**Weaknesses:**

- There are four types of entities in the EDGAR10-Q dataset. They are all about numbers.
- Entity types defined in the paper, such as money, dates, etc, may not belong to named entities referring to guidelines of many research datasets (e.g., OntoNotes 5.0, ACE 2004). It is more accurate to refer to these types as attributes of a person or organization. Although some NER taggers (e.g., stanford ner tagger) also recognize the aforementioned types as named entities.
- The objective of the introduced task is unreasonable. Some entities defined in the paper, such as "time", "money", do not depend on the sentence context. It seems to be facts or results of an event. It seems unreasonable to generate a phrase based on an irrelevant sentence.
It is more accurate to define the task as to generate a concept description given a sentence rather than context NER.
- The contribution of this work is controversial because the quality of the work cannot be judged based on the difficulty it poses for ChatGPT.

**Questions:**

- Is there an annotation file to ensure each instance in the dataset is annotated under the same guideline?
- Why use BERT-Base as the baseline rather than using the FInBERT?

---

> ### Author Response · Authors · 2023-11-23
> **Response to Reviewer uqH6**
>
> **R2.1: "Composition of the dataset"** Thank you for highlighting the composition of entities in the EDGAR10-Q dataset. Indeed, given the nature of finance data, numerical entities play a predominant role, leading to a dataset where the majority of entities are numeric in nature.
> We appreciate your acknowledgment of this inherent characteristic of financial data. If there are specific studies or analyses you recommend to further investigate or enhance the dataset, we are more than willing to consider and pursue them.
>
> **R2.2: "Definition of the entity / Attributes"** Thank you for bringing up the consideration. We appreciate your insight, and to address this concern, we have incorporated a note in the dataset description section of the main paper, acknowledging the distinction between these entity types and those commonly referred to in established research datasets. We believe that this clarification enhances the accuracy and transparency of our work. Furthermore, we are open to any additional suggestions or recommendations on how to address this point in the final draft of the paper.
>
> **R2.3: "Objective of the task"** Thank you for your suggestion regarding the objective of the introduced task in our paper. We have reviewed your comment and have made adjustments to the task definition based on your feedback.
> The revised task is now defined as "to generate a concept description of the entity given a sentence." We believe this modification better aligns with the intention of understanding the meaning of entities within the context of the sentence in which they are present. While we acknowledge that entities like "time" and "money" may convey facts or results of an event, our aim is to delve into the nuanced meanings these entities hold within the specific context of a sentence.
> As an illustration, consider the example provided in Table 1 of the paper: "There were impairments of $0.8 million for the three months ended June 30, 2020 and $2.2 million for the six months ended June 30, 2020." The task allows us to explore the meaning of "$2.2 million" as "Valuation allowance for loan servicing rights with impairments," providing a richer understanding of the entity in the given context.
> We appreciate your valuable input, and if you have any further suggestions or comments, we are open to incorporating them to enhance the clarity and precision of our paper.
>
> **R2.4: "ChatGPT scores"** Thank you for your feedback. We would like to clarify that while ChatGPT is indeed used as a baseline in our study, the primary contributions of our paper lie in the creation of a novel dataset, the introduction of artifact models, and the establishment of rule-based baselines.
> The utilization of ChatGPT as a baseline is intended to highlight that even state-of-the-art models face challenges with the tasks presented in our work. This is meant to provide context and a benchmark for evaluating the performance of our proposed models and baselines.
> We appreciate your attention to the nuances of our contribution, and we welcome any specific suggestions or feedback you may have to further strengthen the paper.
>
> **R2.5: "Annotation Guideline"** Thank you for your queryt. To ensure consistency, all instances in the dataset adhere to the guidelines provided by the SEC filing section for public organizations. These guidelines are outlined in the "Prepare filing documents" section, and we have included a reference to this information in our revised paper.
> For your convenience and reference, the specific SEC guidelines can be found at (https://www.sec.gov/edgar/filer-information/how-do-i). We believe that this additional information enhances the transparency of our annotation process and ensures uniformity in the dataset. Please also check general response G1.1 for clarification. We look forward to addressing any additional points you may have.
>
> **R2.6: "Using FinBERT instead of BERT in baseline approach"** We appreciate your inquiry and have included the results of FinBERT in the appendix of the revised paper for your reference.
> In our experimentation, we found that FinBERT yielded results that were nearly equivalent to those of the vanilla BERT model. The reason behind this similarity is attributed to the fact that the baseline approach  primarily relied on the phrase generation algorithm described in the paper and the MRC model did not significantly impact the baseline scores. The primary responsibility for the baseline scores lay with the phrase extraction algorithm.
>
> We trust that this additional information provides clarity on the choice of baselines in our study. If you have any further questions or suggestions, please do not hesitate to let us know. We are committed to addressing any concerns to enhance the quality and understanding of our work.

---

### Official Review · Reviewer_kwQd · 2023-11-01

**Soundness:** 3 good
**Presentation:** 3 good
**Contribution:** 2 fair
**Rating:** 6
**Confidence:** 4

**Summary:**

This paper introduces the context-ner task of generating relevant context for entities in a sentence and presents the EDGAR10-Q dataset, which is a large dataset with 1M sentences and 2.8M entities. The paper proposes a baseline approach and evaluate various models, where T5-large achieves state-of-the-art results on the context-ner task. Additionally, they found that pre-trained on the context-ner task can help on other finance downstream tasks. Overall, the paper's contributions include introducing a new task and dataset, proposing a baseline approach, and evaluating various models on the task.

**Strengths:**

- This paper introduces a new task of generating relevant context for entities in a sentence, which is a novel problem formulation.The EDGAR10-Q dataset, which is the largest of its kind with 1M sentences and 2.8M entities, is a significant contribution to the field.
- The authors also propose a baseline approach that uses a combination of question generation and reading comprehension to generate contextual phrases for entities.
- The paper is well-written and clearly presents the problem formulation, baseline approach, and evaluation results. The authors provide detailed descriptions of the models evaluated and the evaluation metrics used. The paper also includes a human evaluation case study, which adds to the quality of the paper.
- The new task of generating relevant context for entities in a sentence has interesting applications in finance, and is somewhat surprising for to improve the downstream performance by a large margin. The EDGAR10-Q dataset is a valuable resource for researchers working on similar problems. Overall, this paper makes significant contributions and has the potential to inspire further research in this area.

**Weaknesses:**

- One potential weakness is that the paper does not provide a clear annotation procedure for the dataset. I understand the procedure of collecting publicly available annual reports and extract the paragraphs, but I cannot search any detail about how to get the phrase labels given these entities. Are they automatically extracted from the annual report? Or the labels are annotated by humans following the instruction in Appendix D.1?
- It's unfair to compare the model fine-tuned on EDGAR10-Q with the BloombergGPT-50B model since the model performs the task via a few-shot manner while the T5 model is fine-tuned on the dataset. And a dataset analysis about the potential entity overlap between EDGAR10-Q and the downstream benchmark (e.g., FiQA) would be more helpful to understand the benefits. Also, what would happen if we continually pre-train the model on EDGAR10-Q corpus using T5 objectives (i.e., not predicting the phrase label, but pre-training on these finance text).
- While the authors report the overall performance of the models, it would be much better if they provide a breakdown of the types of errors made or the specific examples where the models fail / be improved after fine-tuning on EDGAR10-Q. This information would be valuable for understanding why the model performance is improved after fine-tuning on EDGAR10-Q.

**Questions:**

See above

**Details Of Ethics Concerns:**

The author proposes a new dataset, which contains public information, so I think they need further legal compliance.

---

> ### Author Response · Authors · 2023-11-23
> **Response to Reviewer kwQd**
>
> **R1.1: "Annotation procedure for the dataset"** Please check general response G1.1.
>
> **R1.2: "Bloomberg GPT baseline and effect of continual pretraining"** Thank you for your comment.
> Regarding the comparison with the BloombergGPT-50B model, we understand your point about the different methodologies employed in our model fine-tuning on EDGAR10-Q and the few-shot manner of BloombergGPT-50B. Our intention in using BloombergGPT-50B as a baseline was to assess the relative performance of our model after prefinetuning with a finance dataset. As stated in our paper, "We use BloombergGPT-50B (2023) 10 shot score as the baseline for these tasks." Our primary objective was to demonstrate the improvement achieved through prefinetuning when compared to a vanilla T5 model.
> To address your suggestion about a dataset analysis on potential entity overlap between EDGAR10-Q and downstream benchmarks such as FiQA, we have conducted an analysis and included the results in the appendix. The analysis indicates that the overlap between the EDGAR10-Q training set and the sentiment analysis tasks of FiQA and Fpb datasets is minimal, nearing 0%.
> Additionally, we appreciate your curiosity about the effects of continual pre-training on the model using the T5 objectives on the EDGAR10-Q corpus. We conducted experiments in this regard, employing continual pre-training with 100K samples from the dataset (referred to as Cont. PT EDGAR T5) followed by fine-tuning on downstream financial tasks. The results, outlined in the appendix, demonstrate that while this approach yielded improved performance compared to a vanilla T5 model, it fell short of the performance achieved with our EDGAR T5 model. We acknowledge that the limitation of not using the entire training split for continual pre-training may have influenced these results, and we thank you for bringing attention to this point.
> We hope these additions and clarifications adequately address your concerns.
>
> **R1.3: "Specific examples where the models fail after fine-tuning on EDGAR10-Q"** We appreciate your suggestion to provide a breakdown of the types of errors made by the models or specific examples where improvement is observed after fine-tuning on EDGAR10-Q.
> In response to your suggestion, we conducted a case study with 100 samples from the test set, specifically highlighting instances where the model exhibited shortcomings, as evidenced by a Rouge-L score of 0. The results of this case study, along with illustrative examples, have been included in the appendix of our paper. We believe that this additional information will contribute to a deeper understanding of the factors influencing the model's performance and the tangible improvements achieved through fine-tuning on the EDGAR10-Q dataset.
> Furthermore, we are open to conducting any additional studies or analyses to enhance the clarity and comprehensibility of our paper. Your insights are invaluable, and we welcome any further guidance to improve the overall quality of our work.

---

### Author Response · Authors · 2023-11-23
**General Response**

We thank reviewers for their insightful feedback.

We address the common concerns in this section and address specific queries in the respective rebuttal sections.

**G1: Annotation process of the dataset**
The gold labels in our dataset are meticulously created by human annotators who are domain experts and possess a comprehensive understanding of the organizations involved. The annotation process follows guidelines from the SEC (U.S. Securities and Exchange Commission) for filing reports. As highlighted on the SEC website (https://www.sec.gov/oiea/investor-alerts-and-bulletins/how-read-10-k10-q?_gl=1*32qw6m*_gcl_au*MTc5ODMzNDg2Ni4xNjk5MDYzOTU1), “U.S. companies are required to present their financial statements according to a set of accounting standards, conventions and rules known as Generally Accepted Accounting Principles, or GAAP. An independent accountant audits the company’s financial statements. For large companies, the independent accountant also reports on a company’s internal controls over financial reporting.”

Importantly, the gold labels are not automatically extracted from the annual reports; rather, they are carefully annotated by human experts following the guidelines and procedures outlined in the SEC regulations. Given the critical nature of these reports in depicting an organization's financial well-being, they are prepared with utmost care, often under the oversight of the management team. As emphasized on the same SEC website,“Laws and regulations prohibit companies from making materially false or misleading statements. Likewise, companies are prohibited from omitting material information that is needed to make the disclosure not misleading. In addition, a company’s CFO and CEO must certify to the accuracy of the 10-K and 10-Q.”

We have added this portion in the appendix and hope that this clarification addresses your concerns regarding the annotation procedure.

**G2: Additional experiments:**

We conduct the following experiments based on reviewers’ suggestions:

**G2.1: Case study about specific examples failing after finetuning T5 with the dataset**  We conducted a case study with 100 samples from the test set, specifically highlighting instances where the model exhibited shortcomings, as evidenced by a Rouge-L score of 0. The results of this case study, along with illustrative examples, have been included in the appendix of our paper.

**G2.2:  Baseline with FinBERT** We have included the results of FinBERT in the appendix of the revised paper for your reference. In our experimentation, we found that FinBERT yielded results that were nearly equivalent to those of the vanilla BERT model.

**G2.3 Effect of continual pre training**  We did continual pre training with 100K samples of the dataset (we call this mode Cont. PT EDGAR T5) and then later fine tuned with the same downstream financial tasks described in the paper. The results were better than vanilla T5 but lesser than EDGAR T5.

---

### Meta-Review · Area_Chair_JMun · 2023-12-12

**Metareview:**

This paper introduces a newly annotated dataset and presents an approach for the task of named entity recognition (NER). The authors propose to generate context for the entities. While the reviewers appreciate the novelty of the idea of generating context for NER, there are several strong reservations from multiple reviewers. Specifically, the data annotation process needs to be better documented, and it is not clear if the current level of detail (even after revision) is sufficient. Other concerns include the experimental setup and the underlying entities involved, which appear to deviate from standard definitions of entities. This deviation raises questions about the significance of the context involved in the recognition process. Overall, it is challenging to address these issues completely in this round of review.

**Justification For Why Not Higher Score:**

There are strong reservations from multiple reviewers on this work.

**Justification For Why Not Lower Score:**

NA

---

### Decision · Program_Chairs · 2024-01-16

Reject